# USING THE VARIATION OF THE GRADIENT TO ACCELERATE FIRST-ORDER OPTIMIZATION ALGORITHMS

## ABSTRACT

Several stochastic optimization algorithms are currently available. In most cases, selecting the best optimizer for a given problem is not an easy task. Therefore, instead of looking for yet another absolute best optimizer, accelerating existing ones according to the context might prove more effective. This paper presents a simple and intuitive technique to accelerate first-order optimization algorithms. When applied to first-order optimization algorithms, it converges much more quickly and achieves lower function/loss values when compared to traditional algorithms. The proposed solution modifies the update rule, based on the variation of the direction of the gradient during training. Several tests were conducted with SGD, AdaGrad, Adam and AMSGrad on three public datasets. Results clearly show that the proposed technique, has the potential to improve the performance of existing optimization algorithms.

## 1 INTRODUCTION

The majority of existing optimization algorithms are first-order methods — use first derivatives of the function to minimize —, based on the gradient descent technique. These techniques (e.g. the back propagation of error) are used to find a matrix of weights that meets the error criterion. Adam (Adaptive Moment estimation) (Kingma & Ba, 2014) is probably the most popular (Goodfellow et al., 2016; Isola et al., 2017). However, Adam (while possessing convergence proof issues) has recently been proven unable to converge to the optimal solution for a simple convex optimization setting (Reddi et al., 2018). AMSGrad addresses this issue by endowing Adam and other adaptive stochastic methods with a long-term memory. Unfortunately, Adam can still outperform AMSGrad in some cases [1]. Therefore, instead of looking for the absolute best optimizer, finding a way to accelerate the existent would be of greater use. In this perspective, we propose a simple technique aiming at improving the empirical performance (find a lower minimum for the loss function) of any first-order optimization algorithm, while preserving their property. We will refer to the solution when applied to an existing algorithm A as AA for (Accelerated Algorithm). For example, for AMSGrad and Adam, the modified versions will be mentioned as AAMSGrad and AAdam. The solution is based on the variation of the direction of the current gradient with respect to the direction of the previous update. We conducted several tests on problems where the shape of the loss is simple (has a convex form) like Logistic regression, but also with non-trivial neural network architectures such as Multi-layer perceptron, deep Convolutional Neural Networks and Long Short-Term Memory. We used *MNIST*, *CIFAR-10* and *IMDB movie reviews* Maas et al. (2011) datasets to validate the solution. Our experiments are limited to SGD, AdaGrad, Adam and AMSGrad.

## 2 RELATED WORK

### 2.1 HOW GRADIENT DESCENT METHOD WORKS ?

Let $f(x)$ be a function parameterized by a model's parameters $x \in \mathbb{R}^n$, sufficiently differentiable of which one seeks a minimum. The gradient method builds a sequence that should — in principle — work towards finding the minimum. For this, we start from any value $x_0$ (i.e. a random value) and

---

[1]https://fdlm.github.io/post/amsgrad/

we construct the recurrent sequence by:

$$x_{t+1} = x_t - \alpha \cdot \nabla f_t(x_t) \tag{1}$$

where $\alpha$ is the learning rate. For adaptive methods like Adam, the learning rate varies for each parameter. This method is ensured to converge even if the input sample is not linearly separable, to a minimum of the error function for a well-chosen learning rate. There exist several variants of this method in two basic categories: first-order and second-order methods. While first-order methods use first derivatives of the function to minimize, second-order methods make use of the estimation of the Hessian matrix (second derivative matrix of the loss function with respect to its parameters) (Schaul et al., 2013; LeCun et al., 1993). The latter determines the optimal learning rates (or step size) to take for solving quadratic problems. While such approach provides additional information useful for optimization, computing accurate second-order derivatives is too computationally expensive for large models. Furthermore, the computed value usually equals to a bad approximation of the Hessian. In this paper, we will only focus on first-order techniques.

## 2.2 EXISTING FIRST ORDER OPTIMIZATION ALGORITHMS

Adam is a first-order gradient-based algorithm of stochastic objective functions, based on adaptive estimates of lower-order moments. The first moment, once normalized by the second moment, gives the direction of the update. Adam updates are directly estimated using a running average of the first and second moment of the gradient. It computes adaptive learning rates for each parameter. In addition to storing an exponentially decaying average of past squared gradients $v_t$, Adam also keeps an exponentially decaying average of past gradients $m_t$, similar to momentum. It has two main components: a momentum component and an adaptive learning rate component. Adam has a bias-correction feature that helps slightly outperform previous adaptive learning rate methods towards the end of optimization when gradients become sparser (Ruder, 2016). To our knowledge, Adam is one of the latest state-of-the-art optimization algorithms used by many practitioners of machine learning. There exist several techniques to improve Adam such as fixing the weight decay (Loshchilov & Hutter, 2017), using the sign of the gradient in distributing learning cases (Bernstein et al., 2018), or switching from Adam to SGD (Keskar & Socher, 2017). However, it has been recently proven that Adam is unable to converge to the optimal solution for a simple convex optimization setting (Reddi et al., 2018). A more recent algorithm (AMSGrad) fixes this problem by endowing the techniques with a long-term memory. The main difference between AMSGrad and ADAM is that AMSGrad maintains the maximum of all $v_t$ (exponentially decaying average of past squared gradients) until the present time step and uses this maximum value for normalizing the running average of the gradient. The original paper claims that AMSGrad either performs similarly, or better than Adam on some commonly used problems in machine learning. The update rule of AMSGrad is as follows :

$$
\begin{aligned}
x_{t+1} &= x_t - \frac{\alpha}{\sqrt{\hat{v}_t + \epsilon}} \hat{m}_t \\
m_t &= \beta_1 \cdot m_{t-1} + (1 - \beta_1) \cdot \nabla f_t(x_t) \\
v_t &= \beta_2 \cdot v_{t-1} + (1 - \beta_2) \cdot \nabla f_t(x_t)^2 \\
\hat{m}_t &= \frac{m_t}{1 - \beta_1^t} \\
\hat{v}_t &= max(\hat{v}_{t-1}, \frac{v_t}{1 - \beta_2^t}).
\end{aligned}
$$

Recall that an optimizer aims to look for parameters that minimize a function, knowing that we do not have any knowledge on what this function looks like. If we knew the shape of the function to minimize, it would then be easy to take accurate steps that lead to the minimum. While we can't plan for the shape, each time we take a step (using any of the optimizers) we can know if we passed a minimum by computing the product of the current and the past gradient, and check whether it is negative. This informs us on the curvature of the loss surface and is — as far as we know — the only accurate information available in real-time. The proposed method exploits this knowledge to improve the convergence of an optimizer. None of the existing techniques uses the variation of the direction of the gradient as information to compute the next update. The technique can be seen as a momentum technique with a condition that specifies when to add momentum.

# 3 A METHOD TO ACCELERATE FIRST-ORDER OPTIMIZATION ALGORITHMS

## 3.1 NOTATION

$\|M_i\|_2$ denotes the $l_2$-norm of $i^{th}$ row of M. The projection operation $\prod_{\mathcal{F},A}(y)$ for A $\in \mathcal{S}_+^d$ (the set of all positive definite $d * d$ matrices) is defined as arg $min_{x \in \mathcal{F}} \left\| A^{1/2}(x-y) \right\|$ for $y \in \mathbb{R}^d$. $\mathcal{F}$ has bounded diameter $D_\infty$ if $\|x-y\|_\infty \leq D_\infty$ forall $x,y \in \mathcal{F}$. All operations applied on vectors or matrices are element-wise.

## 3.2 INTUITION AND PSEUDO-CODE

The intuition behind proposed solution is presented in this section. The pseudo-code of our the method applied to the generic adaptive method setup proposed by Reddi et al. (2018) is illustrated in algorithm 1. For Adam, $\psi_t(g_1, ..., g_t) = (1-\beta_2)\sum_{i=1}^t \beta_2^{t-i}g_i^2$. For AMRSgrad, $\psi_t(g_1, ..., g_t) = (1-\beta_2) \text{ diag}\left(\sum_{i=1}^t \beta_2^{t-i}g_i^2\right)$.

---

**Algorithm 1** *Accelerated - Generic Adaptive Method Setup*

---

**Input:** $x_1 \in$ F, step size $(\alpha_t > 0)_{t=1}^T$, sequence of functions $(\varphi_t, \psi_t)_{t=1}^T$
$t \leftarrow 1$
$S \leftarrow$ threshold
**repeat**
    $g_t = \nabla f_t(x_t)$
    $V_t = \psi_t(g_1, ..., g_t)$
    **if** $g_t \cdot m_{t-1} > 0$ and $\mid m_{t-1} - g_t \mid > S$ **then**
        $gm_t = g_t + m_t$
        $m_t = \beta_1 \cdot m_{t-1} + (1-\beta_1)\dot{g}m_t$
    **else**
        $m_t = \beta_1 \cdot m_{t-1} + (1-\beta_1)\dot{g}_t$
    $\hat{x}_{t+1} = x_t - \alpha_t m_t V_t^{-1/2}$
    $x_{t+1} = \prod_{F,\sqrt{V_t}}(\hat{x}_{t+1})$
    $t \leftarrow t + 1$
**until** t > T

---

Lets take the famous example of the ball rolling down a hill. If we consider that our objective is to bring that ball (parameters of our model) to the lowest possible elevation of the hill (cost function), what the method does is to push the ball with a force higher than the one produced by any optimizer, if the ball is still rolling in the same direction. This is done by adding the sum between the current computed gradient ($g_t$) and the previous gradient ($g_{t-1}$) or the previous moment ($m_{t-1}$) to the step currently being computed instead of only $g_t$. The ball will gain more speed as it continues to go in the same direction and lose its current speed as soon as it will pass over a minimum. To avoid the oscillation of the ball over a minimum, a constraint was added concerning the difference between the past gradient and the current one. This difference should be more than a threshold ($S < \mid g_{t-1} - g_t \mid$) in order for the acceleration to be effective when updates became smaller.

The higher the threshold, the closer we will get to the original algorithm. However, $S$ should not be too small (see section 5.2). Another solution could be to stop accelerating the ball and let the original optimizer take full control of the training once the direction changes for the first time. This solution may also work, but would converge slower. In this paper, we will only focus on the case where the gradient $g_t$ is replaced by $g_t + m_{t-1}$ or $g_t + g_{t-1}$ when computing $m_t$ in Adam and AAmsGrad (see algorithm 1 for details) or $g_t$ in other optimizers (SGD and AdaGrad for example) if the direction does not change. That change we made to the optimizer do not alter its convergence since the quantity $\Gamma_t$ (Reddi et al., 2018) — which essentially measures the change in the inverse of learning rate of the adaptive method with respect to time — will keep its property.

---

**Algorithm 2** *Accelerated - Generic Gradient Descent Method*

---

    **Input:** $x_1 \in$ F, step size $(\alpha > 0)$
    $t \leftarrow 1$
    $S \leftarrow$ threshold
    **repeat**
        $g_t = \nabla f_t(x_t)$
        $new\_g = g_t$
        **if** $g_t \cdot g_{t-1} > 0$ and $\mid g_{t-1} - g_t \mid > S$ **then**
            $new\_g = g_t + g_{t-1}$
        $x_{t+1} = x_t - \alpha_t \cdot new\_g$
        $t \leftarrow t + 1$
    **until** t > T

---

### 3.3 CONVERGENCE ANALYSIS

We assume that if we are able to prove that modifying one optimizer with the proposed method does not alter its convergence, then the same applies for the other optimizers. Thus, we only analyse the convergence of one deterministic optimization method AGD (Accelerated Gradient Descent) and one non-deterministic method (AAMSGrad).

For deterministic methods such as GD, the convergence analysis consists in finding an upper bound of the difference between the function value at the $t$-th iteration and the minimal value $(f(x_t) - f(x^*))$, where f is the function to minimize, $x^*$ is the optimal solution and $x_t$ is the solution found by the algorithm at time $t$.

**Theorem 1.** *Let $f : \mathbb{R}^d \to \mathbb{R}$ be a L-Lipschitz convex function and $x^* = argmin_x f(x)$. Then, GD with step-size $\alpha \leq 1/L$ satisfies the following :*

$$\sum_{t=0}^{T-1}(f(x_{t+1}) - f(x^*)) \leq \frac{L}{2}[\|x_0 - x^*\|^2 - \|x_T - x^*\|^2] \tag{2}$$

$$f(x_T) - f(x^*) \leq \frac{L}{2T}\|x_0 - x^*\|^2$$

*In particular, $\frac{L}{\epsilon}\|x_0 - x^*\|^2$ iterations suffice to find an $\epsilon$-approximate optimal value x. GD has a sublinear convergence rate of $O(1/T)$.*

For AGD, since $\nabla f(x_{T-1}) = k\nabla f(x_T)$ if $\nabla f(x_{T-1}) \cdot \nabla f(x_T) > 0$ with $k \geq 1$ we have :

**Theorem 2.** *Let $f : \mathbb{R}^d \to \mathbb{R}$ be a L-Lipschitz convex function and $x^* = argmin_x f(x)$. Then, AGD with step-size $\alpha \leq 1/L$ satisfies the following :*

$$f(x_T) - f(x^*) \leq \frac{L(1+k)}{2T}\|x_0 - x^*\|^2 \tag{3}$$

*AGD has a sub-linear convergence rate of $O(1/T)$ which is the same as GD. The method does not alter the convergence of the algorithm.*

For AAMSGrad, the analysis differs slightly from GD (deterministic methods) but the underlying goal remains the same. We used the online learning framework (Zinkevich, 2003). As such, the convergence result for AAMSGrad is taken with respect to the *regret* since expected values of the gradients of the stochastic objective are difficult to compute. The *regret* is defined as :

$$R(T) = \sum_{t=1}^{T}[f_t(x_t) - f_t(x^*)] \tag{4}$$

**Theorem 3.** *Assume that $\mathcal{F}$ has bounded diameter $D_\infty$ and $\|\nabla f_t(x)\|_\infty \leq G_\infty$ for all $t \in [T]$ and $x \in \mathcal{F}$. With $\alpha_t = \frac{\alpha}{\sqrt{T}}$, AAMSGrad achieves the following guarantee, for all $T \geq 1$ :*

$$R_T \leq \frac{D_\infty^2 \sqrt{T}}{2\alpha(1-\beta_1)} \sum_{i=1}^{d} (\hat{v}_{T,i}^{-1/2}) + \frac{D_\infty^2}{4(1-\beta_1)} \sum_{t=1}^{T} \sum_{i=1}^{d} \frac{\beta_{1t} \hat{v}_{t,i}^{1/2}}{\alpha_t}$$

$$+ \frac{2\alpha\sqrt{1+log(T)}}{(1-\beta_1)^2(1-\gamma)\sqrt{(1-\beta_2)}} \sum_{i=1}^{d} \|g_{1:T,i}\|_2 \tag{5}$$

The proof of this bound is given in the appendix. The following result falls as an immediate corollary of the above result :

**Corollary 3.1.** *Let $\beta_{1t} = \beta_1 \lambda^{t-1}, \lambda \in (0,1)$ in Theorem 2, then we have :*

$$R_T \leq \frac{D_\infty^2 \sqrt{T}}{2\alpha(1-\beta_1)} \sum_{i=1}^{d} (\hat{v}_{T,i}^{-1/2}) + \frac{\beta_1 D_\infty^2 G_\infty}{4(1-\beta_1)(1-\lambda)^2}$$

$$+ \frac{2\alpha\sqrt{1+log(T)}}{(1-\beta_1)^2(1-\gamma)\sqrt{(1-\beta_2)}} \sum_{i=1}^{d} \|g_{1:T,i}\|_2 \tag{6}$$

The *regret* of $AAMSGRAD$ can be bounded by $O(2G_\infty\sqrt{T}) \simeq O(G_\infty\sqrt{T})$. The term $\sum_{i=1}^{d} \|g_{1:T,i}\|_2$ can also be bound by $O(G_\infty\sqrt{T})$ since $\sum_{i=1}^{d} \|g_{1:T,i}\|_2 << dG_\infty\sqrt{T}$ (Kingma & Ba, 2014).

## 4 EXPERIMENTS

We investigated different models: Logistic Regression, Multilayer perceptron (MLP), Convolutional Neural Networks (CNN) and Long Short-Term Memory (LSTM). We used the same parameter initialization for all of the above models. $\beta_1$ was set to $0.9$ and $\beta_2$ was set to $0.999$ as recommended for Adam (Kingma & Ba, 2014) and the learning rate was set to $0.001$ for Adam, AAdam, AMSGrad, AAMSGrad and $0.01$ for the remaining optimizers. The algorithms were implemented using Keras and the experiments were performed using the built-in Keras models, making only small edits to the default settings. For each model, all the optimizers started with the same initial values of weights. For the accelerated versions AAdam and AAMSGrad, we will only show the variant where the value of $m_t$ was changed according to algorithm 1. However, both solutions ($m_{t-1} + g_t$ or $g_{t-1} + g_t$) worked well during our experiments. The experiments were repeated 10 times and we calculated the mean of the estimated mean model skill. Thus, each point in the following figures represents the average result after 10 trainings on one epoch (one pass over the entire dataset). The number of epochs varied from 10 to 30 for all experiments in this paper (there is a good chance for models to over fit with a higher number of epochs). For AMSGrad, we only considered the version with 'max' on $v_t$, not the diagonal. The full code for this project is publicly available (tensorflow and keras) [2].

### 4.1 BASICS PROBLEMS

We first tested the proposed method on two basics functions: a convex function $x^2$ (with the minimum fixed at 0) and a non-convex function $x^3$ (minimum $\rightarrow -\infty$). As seen in figure 1, the accelerated versions are able to reach a lower loss function value compared to the originals in both cases.

### 4.2 IMAGE RECOGNITION

No pre-processing was applied to MNIST images. All optimizers were trained with a mini-batch size of 128. All weights were initialized from values truncated using normal distribution with a standard deviation of 0.1. We used the 'softmax cross entropy' as the activation function. Again, for AAMSGrad we used $g_{t-1}$ instead of $m_{t-1}$ in the if-else condition and the computation of the

---

[2]http://tiny.cc/g0nddz

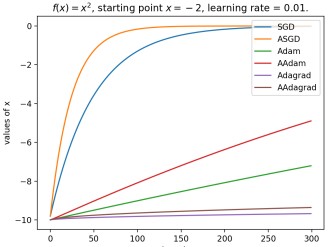 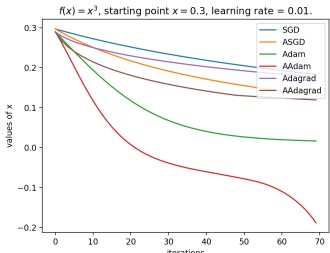

Figure 1: Minimizing basics functions with starting points x = -10 for $x^2$ and x = 0.3 for $x^3$. For AAdam, we used $g_{t-1}$ instead of $m_{t-1}$ in the if-else condition and the computation of the new step. No threshold was set.

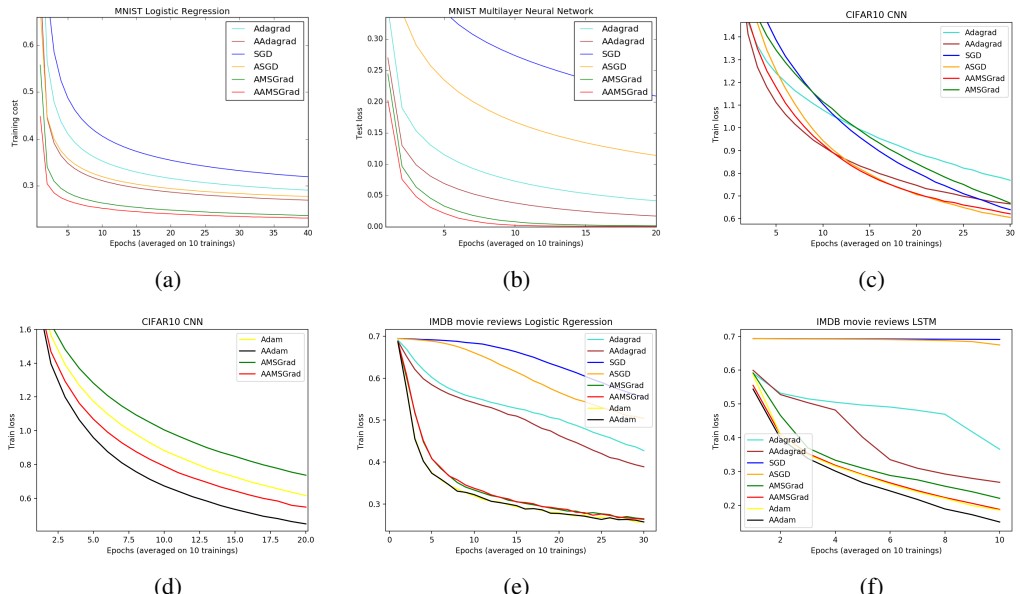

Figure 2: Logistic regression training negative log likelihood on MNIST images (a). MLP on MNIST images (b). Loss on training step. Convolutional neural networks on cifar10. Training cost over 30 (c) and 20 epochs (d). CIFAR-10 with c32-c32-c64-c64 (c) and c32-c64-c128-c128 (d) architectures. Logistic Regression training negative log likelihood on IMDB movie reviews (e). LSTM training negative log likelihood on IMDB movie reviews (f). Loss on training step. No threshold was set when training the LSTM model.

new step. Logistic regression has a well-studied convex objective, making it suitable for comparison of different optimizers without worrying about local minimum issues. We implemented the same model used for comparing Adam with other optimizers in the original paper on Adam. In figure 2a, we can see that the accelerated versions outperform the original algorithms from the beginning to the end of the training.

Multi-layer perceptron (MLP) is a powerful model with non-convex objective functions that consists in an input layer, an output layer, and $n$ hidden layers where n $>= 1$. In our experiments, we selected models that were consistent with previous publications in the area. A simple neural network model with one fully connected hidden layer with 512 hidden units and ReLU activation was used for this experiment with a mini-batch size of 128. Figure 2b shows the results of running a MLP on MNIST data. Again, the accelerated versions outperform (reach a lower minimum) the original algorithms on the training step.

### 4.3 CIFAR-10 DATASET

CNNs (LeCun et al., 1995) are neural networks where layers represent convolving filters (Krizhevsky et al., 2012) applied to local features. CNNs are now used in almost every task in machine learning (i.e. speech analysis (Abdel-Hamid et al., 2014), text classification (Tato et al., 2017), etc.). We considered the multi-class classification problem on the standard CIFAR-10 dataset, which consists of 60,000 labelled examples of $32 \times 32$ images. We used a CNN with several layers of convolution, pooling and non-linear units. In particular, the first architecture (which results are shown in figure 2c), contains 4 convolutional layers with 32 channels for the first two layers and 64 channels for the other two. The kernel of each of the 4 convolutional layers has a size of $3 \times 3$ followed by one fully connected layer of size 212. The network uses $2 \times 2$ max pooling. It also contains three dropout layers with keep probabilities of $0.25$, $0.25$ and $0.5$, respectively applied 1) between the two first convolutional layers and the two other convolutional layers; 2) between the two last convolutional layers and the flatten layer; and 3) between the fully connected layers. The mini-batch size is set to 128 like in previous experiments. The code for this architecture [3] was taken from Keras. Results for this problem are reported in figures 2c and 2d. As we can see, accelerated algorithms performed considerably better than originals on the two different architectures.

### 4.4 IMDB MOVIE REVIEWS DATASET

Finally, we experimentally evaluated our method on imdb movie reviews data [4]. This dataset consists of 25,000 movies reviews from IMDB, labelled by sentiment (positive/negative). Reviews have been pre-processed and encoded as a sequence of word indexes. We then ran a Logistic Regression and LSTM models. LSTM has become a core component of neural Natural Language Processing (NLP) (Tai et al., 2015). The LSTM architecture (Hochreiter & Schmidhuber, 1997) can learn long-term dependencies using a memory cell that is able to preserve states over long periods of time. We used the built-in LSTM architecture of Keras with 32 units, a recurrent dropout of $0.2$ and a dropout of $0.2$. An embedding layer preceded the LSTM. The dimensionality of character embedding was set to 32 and the batch size was set to 32. We took the first 5,000 most frequent words. Results are presented in figures 2e and 2f. As expected, accelerated versions consistently outperformed original algorithms. Nevertheless, for Adam and AMSGrad, the results were not very clear from logistic regression (2e), which may be due to the sparse nature of the data. More analyses on sparse problems should be considered in the future.

## 5 DISCUSSION

### 5.1 ANALYSIS OF THE EXECUTION TIME

The proposed method add to the original algorithms, a test (if-else) which might slow the execution time. What if we gave the same execution time to all algorithms, investigating whether the proposed method allows the algorithm to reach a lower minimum? In figure 3, we performed two experiments where we evaluated the minimum reach by each optimizer given a fixed time. For function $x^2$, we set the starting point at $x = 2$ and the time at $t = 0.0005s$. As we can see (Figure 3 - left side), the accelerated versions did less iterations but reached a lower function/loss values when compared to the original algorithms under the same execution time. Moreover, we ran the logistic regression on the MINST dataset with a fixed time of $t = 10s$. Again, as we can see (Figure 3 - right side) accelerated versions largely outperformed the original algorithms. Thus, even if the method takes time compared to original algorithms, it is able to reach lower loss values given the same time of execution.

### 5.2 HOW TO CHOOSE THE THRESHOLD ?

The choice of the parameter $S$ is very important to ensure the effectiveness of the approach. In Figure 4, we tested different values of $S$ with LR on MNIST. As we can see, when the value is high, the accelerated versions seem to behave like the original optimizers. When the value is too small,

---

[3]https://github.com/keras-team/keras/blob/master/examples/cifar10_cnn.py
[4]https://keras.io/datasets/#imdb-movie-reviews-sentiment-classification

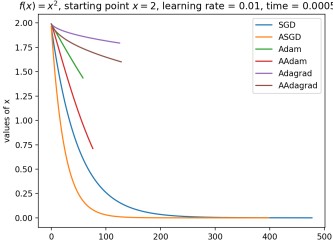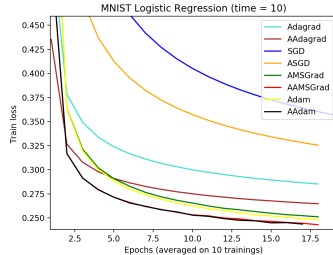

Figure 3: Convergence analysis with a fixed time.

the accelerated versions converge faster at the beginning but fail to reach a better minimum. Note that, for AAMSGrad, $S$ should be smaller than the one set for AAdam since the $v_t$ used to compute the step size in AMSGrad is higher (the max value) than in Adam. We would like to mention that during all our experiments with SGD and AdaGrad we did not set a value for $S$. $S$ was only set for AAdam and AAMSGrad.

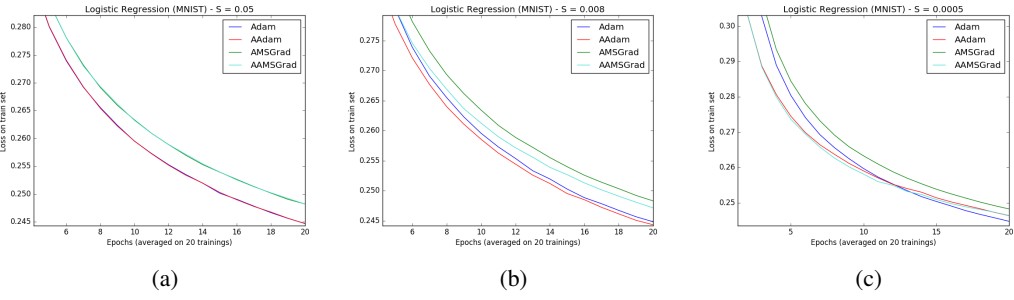

Figure 4: Experiments on the variation of the value of $S$

## 6 CONCLUSION

In this paper, we have presented a simple and intuitive method that modifies any optimizers update rule to improve its convergence. There can be no optimizer working best for all problems. Each optimizer has its advantages and disadvantages. No one can tell in advance which will be the best choice for a specific case. The solution we have presented has the potential to speed up the convergence of any first-order optimizer based on the variation of the direction of the gradient or the first moment ($m_t$ for moving average methods). Instead of using the gradient as is for computing the next step, we used the sum of the current gradient and the past gradient (similar to momentum techniques) to compute the current step when both values have the same sign (the direction did not change), and keep the current gradient otherwise. We conducted successful experiments with four well-known optimizers (SGD, AdaGrad, Adam and AMSGrad) on different models using three public datasets (MNIST, CIFRA-10 and IMDB movies reviews). In all our experiments, the accelerated versions outperformed the originals. In worst cases, both had the same convergence, which suggests that the accelerated versions were at least as good as the originals.

This work took those optimizers one-step further and improved their convergence without noticeably increasing complexity. The only drawback of the proposed solution is that it requires slightly more computation than the standard approach (as it has to compute the if-else instruction), and implies to specify a new parameter $S$. However, we found that the extra time taken by the technique was compensated by the better minimum reached. The new update rule depends only on the variation of the direction of the gradient, which means that it can be used in any other first-order optimizer for the same goal. The technique is intuitive and straightforward to implement.

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

# A APPENDIX

## A.1 PROOF OF THEOREM 3

The proof presented below is along the lines of the Theorem 4 in Reddi et al. (2018), which provides a proof of convergence for AMSGrad.

The goal is to prove that, the *regret* of AAMSGrad is bounded by :

$$R_T \leq \frac{D_\infty^2 \sqrt{T}}{2\alpha(1-\beta_1)} \sum_{i=1}^{d} (\hat{v}_{T,i}^{-1/2}) + \frac{D_\infty^2}{4(1-\beta_1)} \sum_{t=1}^{T} \sum_{i=1}^{d} \frac{\beta_{1t}\hat{v}_{t,i}^{1/2}}{\alpha_t}$$
$$+ \frac{2\alpha\sqrt{1+log(T)}}{(1-\beta_1)^2(1-\gamma)\sqrt{(1-\beta_2)}} \sum_{i=1}^{d} \|g_{1:T,i}\|_2 \tag{7}$$

*Proof.* For each optimization method, we have :

$$x_{t+1} = x_t - \alpha U \tag{8}$$

Where $\alpha$ is the step size. Note that, the value of the update U is the gradient (or slope) of a line. For example, this line is the slope of the tangent of the loss at $x_t$ in SGD.

AAMSGrad has 2 rules for the update which are different depending on whether the direction of the update changes or not. The current step taken by AAMSGrad is :

$$\text{(a) } x_{t+1} = \Pi_{F,\sqrt{\hat{V}_t}}(x_t - \alpha_t m_x/\sqrt{\hat{v}_t}) \qquad \text{si } g_t \cdot m_{t-1} > 0$$
$$\leq \Pi_{F,\sqrt{\hat{V}_t}}(x_t - 2\alpha_t \cdot |m_t|/\sqrt{\hat{v}_t})$$
$$m_x = \beta_1 \cdot m_{t-1} + (1-\beta_1) \cdot (\nabla_{\theta_t} J(\theta) + m_{t-1}) \tag{9}$$
$$\text{(b) } x_{t+1} = \Pi_{F,\sqrt{\hat{V}_t}}(x_t - \alpha_t m_t/\sqrt{\hat{v}_t}) \qquad \text{sinon}$$

$$|(1-\beta_1)m_{t-1}| \leq |\beta_1 m_{t-1} + (1-\beta_1)g_t| \tag{10}$$

Which lead us to:

$$m_x = \beta_1 m_{t-1} + (1-\beta_1)g_t + (1-\beta_1)m_{t-1}$$
$$m_x \leq |2 \cdot (\beta_1 m_{t-1} + (1-\beta_1)g_t)| \tag{11}$$
$$m_x \leq |2 \cdot m_t|$$

All operations are element wise. When the direction of the current update changes from the past one, the current update is the same as in AMSGrad (rule (b)). When the direction stays the same, the update is equal to the sum between the update that AMSGrad would have taken and the value $(1-\beta_1)m_{t-1}$ (rule (2)). Thus the *regret* bound of AAMSGrad is :

$$R_T \leq max(R_{T(a)}, R_{T(b)}) \tag{12}$$

Where $R_{T(b)}$ is the *regret* when we consider only the second update rule of AAMSGrad. Please note that, the bound of $R_{T(b)}$ is similar to that of AMSGrad. $R_{T(a)}$ is the *regret* if we consider only the first rule. The proof will consist of finding the bound for $R_{T(a)}$. The second update rule of AAMSGrad can be rewritten as follows :

$$x_{t+1} \leq \Pi_{F,\sqrt{\hat{V}_t}}(x_t - U) \text{ where } U = 2 \cdot \alpha_t \hat{V}_t^{-1/2} m_t \tag{13}$$

If we ignore the value 2, the proof will be the same as proving the convergence of AMSGrad. Thus, we will not rewrite the demonstration but it is easy to see that, after doing the same steps as Sashank J et al, we have :

$$R_{T(a)} \leq \frac{D_\infty^2}{2(1-\beta_1)} \sum_{i=1}^{d} (\frac{\hat{v}_{T,i}^{-1/2}}{\alpha_T} + \frac{\hat{v}_{2,i}^{-1/2}}{\alpha_2}) + \frac{D_\infty^2}{4(1-\beta_1)} \sum_{t=1}^{T} \sum_{i=1}^{d} \frac{\beta_{1t} \hat{v}_{t,i}^{1/2}}{\alpha_t}$$
$$+ \frac{2\alpha\sqrt{1+log(T)}}{(1-\beta_1)^2(1-\gamma)\sqrt{(1-\beta_2)}} \sum_{i=1}^{d} \|g_{1:T,i}\|_2$$
$$\leq \frac{D_\infty^2\sqrt{T}}{2\alpha(1-\beta_1)} \sum_{i=1}^{d} (\hat{v}_{T,i}^{-1/2}) + \frac{D_\infty^2}{4(1-\beta_1)} \sum_{t=1}^{T} \sum_{i=1}^{d} \frac{\beta_{1t} \hat{v}_{t,i}^{1/2}}{\alpha_t}$$
$$+ \frac{2\alpha\sqrt{1+log(T)}}{(1-\beta_1)^2(1-\gamma)\sqrt{(1-\beta_2)}} \sum_{i=1}^{d} \|g_{1:T,i}\|_2 \tag{14}$$

Because $R_{T(b)} \leq R_{T(a)}$, the *regret* bound of AAMSGrad is :

$$R_T \leq max(R_{T(a)}, R_{T(b)}) = R_{T(a)} \tag{15}$$

$\square$

Which complete the proof.

## A.2  ADAM VS AADAM

Depending on the starting point of the algorithm, Adam can reach a minimum that is not optimal or oscillate around the minimum (as shown in Figure 5 - left side). This is principally due to the fact that $\Gamma_{t+1}$ can potentially be indefinite for t $\in$ [T] Reddi et al. (2018) with:

$$\Gamma_{t+1} = \left( \frac{\sqrt{V_{t+1}}}{\alpha_{t+1}} - \frac{\sqrt{V_t}}{\alpha_t} \right) \tag{16}$$

However, this quantity remains $\leq 0$ for non-exponential moving average algorithms such as SGD or AdaGrad (to be observed that AMSGrad resolves this specific problem). Since the technique amplified the step taken by the original algorithm, we could not expect more than for AAdam to behave like Adam (as shown in Figure 5). Thus, depending on the starting point, AAdam may reach a better minimum faster, or only behave like Adam.

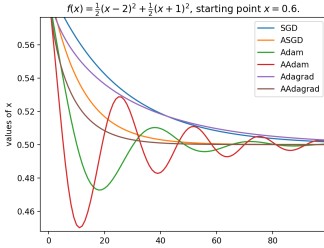 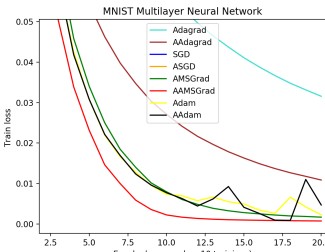

Figure 5: Adam vs AAdam

Overall, the proposed technique can be easily applied to other kinds of optimizers. For example, for NAG (Nesterov Accelerated Gradient) we will have (all operations are element-wise):

$$x_{t+1} = y_t - \alpha \cdot (\nabla f_t(y_t) + \nabla f_{t-1}(y_{t-1}))$$
$$\text{if} \quad \nabla f_t(y_t) \cdot \nabla f_{t-1}(y_{t-1}) > 0 \tag{17}$$
$$y_{t+1} = x_{t+1} + \beta(x_{t+1} - x_t)$$

