# OpenReview forum: "Accelerating First-Order Optimization Algorithms"
_ICLR.cc/2020/Conference — Reject_

### Official Review · AnonReviewer1 · 2019-10-22
**Official Blind Review #1**

**Rating:** 3

**Review:**

The paper describes a technique to speed up optimizers that rely on gradient
information to find the optimum value of a function. The authors describe and
justify their method and show its promise in an empirical evaluation.

The proposed method sounds interesting and promising, but the empirical
evaluation is unclear. In particular, details are missing on the exact
experimental setup and some of the presented results are unconvincing. I refer
to the results of the basic function optimization (Figure 1), which shows that
several of the considered optimizers are unable to even get close to the optimum
of x^2 after several hundred iterations. It seems that this is extremely easy
function to optimize -- why are the considered optimizers performing so poorly
on it? How were the hyperparameters of the optimizers set? This presumably
affects the other results presented in the paper as well, and puts the
improvement of the proposed method in question.


**Experience Assessment:**

I have read many papers in this area.

**Review Assessment: Checking Correctness Of Derivations And Theory:**

I assessed the sensibility of the derivations and theory.

**Review Assessment: Checking Correctness Of Experiments:**

I assessed the sensibility of the experiments.

**Review Assessment: Thoroughness In Paper Reading:**

I read the paper at least twice and used my best judgement in assessing the paper.

---

### Official Review · AnonReviewer2 · 2019-10-23
**Official Blind Review #2**

**Rating:** 3

**Review:**

The authors present an acceleration technique for first-order optimization algorithms by comparing the directions of gradients in consecutive steps, which works for SGD, Adam, and AMSGrad.  Empirically it seems to work well with some standard evaluations with CNN for MNIST and CIFAR10 and LSTM for IMDB, beating the non-accelerated versions in convergence speed. However there are some issues with the parameter choice and proofs. Below are my specific comments:

1. Setting the parameter S seems to be difficult and problem-dependent. S controls the size of the region near the optimum where the algorithm falls back to the non-accelerated version. But S depends on the size of the gradient, which is problem-dependent. If we need to tune S for the algorithm to work well on a particular dataset, then it defeats the purpose of acceleration in the first place.

2. The setting of S also depends on batch size if mini-batch stochastic gradient algorithms are used. In the update rules S is compared against |g_t-1 - g_t|, and this quantity is directly related to the variance of gradients, which in term depends on the batch size. This makes it even more difficult to set a priori.

3. What is k in Theorem 2? In the line above Theorem 2, why is it the case that the gradient at x_T-1 is k times the gradient at x_T? Also, if we compare equations 2 and 3, the regret bound for the `accelerated' version is k times worse than the original non-accelerated SGD. How could this happen?

4. In the proofs in the Appendix I see no mention of the parameter S, which is very strange since it is part of the update condition. The size of S affects the convergence, as shown in Figure 4. It is odd to have a regret bound in Theorem 3 that is completely independent of S.

Unless the authors can address these issues I don't think the current paper is suitable for publication yet.






**Experience Assessment:**

I have read many papers in this area.

**Review Assessment: Checking Correctness Of Derivations And Theory:**

I assessed the sensibility of the derivations and theory.

**Review Assessment: Checking Correctness Of Experiments:**

I assessed the sensibility of the experiments.

**Review Assessment: Thoroughness In Paper Reading:**

I read the paper thoroughly.

---

### Official Review · AnonReviewer3 · 2019-10-23
**Official Blind Review #3**

**Rating:** 1

**Review:**

Summary:

This paper presents an adaptive method which can be used alongside existing accelerated gradient methods. The paper is difficult to read due to mistakes and poorly defined mathematical notation. I believe that the paper is missing reference to related methods. The theoretical analysis in the paper is difficult to follow and provides little insight into the benefits of the proposed approach.

Overview:

There are many mistakes throughout the paper which have made it difficult to read.

Overall, I felt that this paper was missing a discussion of the effect of stochasticity on the proposed method. The issue with measuring the variation in the gradient direction is that in regimes where the gradient noise is dominating the signal the gradient direction at each time step is poorly correlated with overall optimization progress --- thus it seems intuitively ineffective to rely on the gradient direction to adjust the algorithm.

1)  At the bottom of page 2, the authors write "knowing that we do not have any knowledge of what this function looks like". While minor, I would point out that we are able to compute local statistics of the function and so certainly we have _some_ knowledge.

2) The authors claim that no techniques exist which use the variation of the direction of the gradient. One such example is in [1] which uses (in one case) the variation of the gradient direction to determine and appropriate time to restart the momentum computation.

3) In section 3.2, the Adam moment computation is missing a "diag". Assuming that AMRSGrad is AMSGrad (mistyped), then this term is incorrect and matches Adam.

4) There are many mistakes in the Algorithm 1 box.

- The wrong $F$ is used in the input (should be $\mathcal{F}$).
- The algorithm takes as input a sequence of functions ($\phi, \psi$) which are not used.
- Within the if statement, $gm_t = g_t + m_t$. I believe this should be an $m_{t-1}$. It is not clear what the vector $gm$ is exactly, and then $\dot{g}$ is used afterwards which is also not defined.
- The algorithm checks for $|m_{t-1} - g_t| > S$ while the text uses $|g_{t-1} - g_t| > S$.

5) The first line of section 3.3 is quite worrying: "We assume that if we are able to prove that modifying one optimizer with the proposed method does not alter its convergence, then the same applies for the other optimizers". This seems like a dangerous assumption to make and should at the very least be carefully verified empirically. Following this, I am not sure what the authors mean by "deterministic" and "non-deterministic" methods.

6) I do not understand the claim above Theorem 2 that $\nabla f(x_{T-1}) = k \nabla f(x_T)$. Under what conditions does this hold and how is $k$ computed? If I understand correctly, the bound provided in Theorem 2 is worse than that given for gradient descent. Moreover, the bound does not depend on the hyperparameter $S$ introduced in Algorithm 1 and provides limited insights into the method. I could not find a proof of Theorem 2 in the paper or appendix.

7) There are serious flaws with the experimental evaluation in this paper.

a) There is no tuning over hyperparameter settings for any of the optimizers.

b) The basic problems are very limited, even for toy problems. The 1D deterministic quadratic tells us very little about the performance of the optimizer. And the 1D cubic problem is particularly confusing. Unless I am mistaken, the gradient will always have the same sign (3x^2) and thus the acceleration condition will never be triggered.

c) I believe that Figure 2 explores stochastic optimization problems which as discussed at top is a crucial evaluation. Unfortunately, due to lack of parameter tuning it is difficult to infer much about the comparison between the methods.

d) Figure 4 compares performance variation over changing the threshold. The y-axis scale across each plot changes making the comparison unnecessarily difficult --- the scale should be the same.

Minor:

- TYPO Line 2, "minimize ---,"
- End of intro, MNIST and CIFAR not cited while IMDB is. Citation uses citet not citep.
- Bottom of page 2, ""
- Top of section 3.2, "The pseudo code of our the method"

References:

[1] Adaptive Restart for Accelerated Gradient Schemes, Brendan O'Donoghue and Emmanuel Candes



**Experience Assessment:**

I have published one or two papers in this area.

**Review Assessment: Checking Correctness Of Derivations And Theory:**

I assessed the sensibility of the derivations and theory.

**Review Assessment: Checking Correctness Of Experiments:**

I assessed the sensibility of the experiments.

**Review Assessment: Thoroughness In Paper Reading:**

I read the paper thoroughly.

---

### Decision · Program_Chairs · 2019-12-19

**Decision:**

Reject

**Comment:**

All reviewers recommend rejection, and the authors have not provided a response.